# Mandible Angle Resection with the Retroauricular Approach

**DOI:** 10.3390/jcm12072641

**Published:** 2023-04-01

**Authors:** Yoon Joo Lee, Yunsung Park, Yooseok Ha, Sunje Kim

**Affiliations:** 1Doctorsmi Aesthetic Plastic Surgical Clinic, Daejeon 35230, Republic of Korea; 2Department of Plastic and Reconstructive Surgery, College of Medicine, Chungnam National University, Daejeon 34134, Republic of Korea

**Keywords:** retroauricular, mandible, contouring

## Abstract

Square-shaped and large moon-shaped faces are commonly observed in Asians, and the contour of the mandible is associated with the shape of the lower part of the face. Mandible contouring surgery is performed to create a softer impression for East Asians. Currently, most surgeries are performed using an intraoral approach. External approaches have not been cosmetically attempted because of possible damage to the facial nerve and visible scarring and have been limited to mandible bone fracture reduction. This study included 42 patients who underwent mandibular angle reduction via classical intraoral incision and retroauricular incision between April 2019 and October 2021. Clinical outcomes were assessed using the Global Aesthetic Improvement Scale and Visual Analog Scale. Surgery was successful in all cases, with no significant complications. An appropriate mandibular contour was achieved postoperatively. All patients were satisfied with the outcome. Some patients experienced short-term complications, such as hematoma and wound disruption of the skin above the incision line. However, these improved within 3 weeks, and no serious long-term complications were observed. Mandible angle resection with the retroauricular approach is a promising alternative for patients, allowing speedy recovery and the resumption of routine daily life.

## 1. Introduction

The facial structure of Asians is markedly different from that of Caucasians in several aspects, including the skeletal framework. Since the facial structure of Caucasians is relatively dolichocephalic, the width of the mandible must be moderately wide to appear balanced and attractive. Asian faces are usually mesocephalic; therefore, if the mandible is large and square, it provides a relatively hard impression. A wide and square face is considered to be unaesthetic and masculine in Asian cultures, where facial beauty is characterized by an oval appearance. Additionally, with economic development, personal interest and investment in beauty and external appearance have increased, and square-jaw surgery in Asians has become very common. While the cause of square jaws may vary, the most common cause is the mass hypertrophy of the muscles and the hypertrophy of the mandible itself. For Caucasians, surgical treatment for the muscles was devised first, focusing on the development of the square jaw due to the hypertrophy of the masticatory muscles [1]. Furthermore, subsequent treatment methods have predominantly focused on correcting the atrophy of the affected muscles using botulinum toxin injections [2]. For Asians, while hypertrophy of the masseter muscles is evident, the shape of the square contour is considered more prominent due to the skeleton of the mandible bone itself. Therefore, treatment and research have focused more on the analysis of the bony frame and surgical approaches, such as osteotomy [3,4].

There are various methods of mandibular angle resection, such as ostectomy [3], one-stage long curved ostectomy [5], and lateral cortex ostectomy [6]. Most reported surgeries have been performed with an intraoral approach, which has the advantage of avoiding damage to the mandibular branch of the facial nerve; however, several problems have been reported. When the medial pterygoid muscle is detached, the field of vision cannot be secured. Therefore, a blunt dissection must be performed. During this procedure, complications, such as massive bleeding and hemostasis, will occur if the facial artery or mandibular vein is damaged. Occasionally, this procedure has been linked to serious emergency situations, such as airway obstruction caused by edema and massive bleeding. Additionally, after surgery, patients may often complain of discomfort, such as pain resulting from the perioperative opening of their mouth by the operator in order to view the surgical field or dietary restrictions due to intraoral incision and drainage.

The external approach has not been attempted cosmetically due to possible damage to the facial nerve and visible scarring. Moreover, its use has been limited to mandible bone fracture reduction [7]. While mandible angle contouring using Lee’s rasp and saw as a direct approach through the intraoral incision with submental stab incisions has been reported, this method is not widely used [8].

Based on insights from our previous research of mandible reduction using face lifting incisions [9], we performed mandibular angle reduction through a small incision in the retroauricular groove in patients with square-shaped faces, with satisfactory results regarding reduced pain and discomfort and shorter recovery time to routine life. In this study, we introduce a mandible contouring technique using a novel type of incision.

## 2. Materials and Methods

This prospective study compared the surgical outcomes of 42 Asian patients (37 women and 5 men) who underwent mandibular angle reduction between April 2019 and October 2021. The patients were divided into two groups based on whether they underwent surgery by the conventional intraoral or retroauricular incision methods (n = 21 for each group). The mean age of the patients was 34 years, and those with concomitant severe medical illnesses were excluded. The study protocol was reviewed and approved by the institutional review board (approval no. 2020-2801-01). Informed consent was obtained from all participants before study enrollment. A preoperative computed tomography scan was systematically performed to map the course of the inferior alveolar nerve and identify any abnormality or anatomical variation of the mandibular lesion.

### 2.1. Surgical Techniques

All surgeries were performed under general anesthesia by orotracheal intubation. The head was rotated to the side opposite to the operative site. To reduce intraoperative bleeding, surgery was performed approximately 10 min after the administration of a 1:100,000 adrenaline solution on the retroauricular incision line. Surgery using the conventional intraoral incision was carried out as follows. Briefly, vestibular incision and subperiosteal dissection from the anterior ramus to the vestibule mucosa of the premolar were performed. After sufficiently dissecting the posterior border and mandibular angle of the inferior border of the ramus, a fracture was created using an oscillating saw. After hemostasis, the surgical site was closed with absorbable sutures. Surgery using a retroauricular incision was conducted as follows. An incision of approximately 3 cm was made along the retroauricular groove. Key point sutures were made to prevent a prolonged incision during traction (Figure 1). Then, dissection was performed along the hatched area to the subcutaneous plane. The superficial musculoaponeurotic system (SMAS) flap was lifted above the mandibular angular borderline 3 cm below the earlobe, and the contour of the lower parotid gland was examined to prevent iatrogenic damage (Figure 2). After a visual examination of the facial nerve of the premasseteric fascia, attention was paid to nerve damage, and a horizontal division line of approximately 2 cm was made on the masseter fascia between the buccal and upper mandibular branches of the facial nerve (Figure 3). The mandibular angle was approached by splitting it with a small metzembaum scissor in the direction vertical to the masseteric fascia and masseter muscle fibers. The exact location of the incision varied according to the location of the parotid gland and facial nerve. After sufficient subperiosteal dissection, the cutting line on the bone was marked with a pencil, and mandibular contouring was performed along the line using a saw (Figure 4 and Figure 5). At this time, the assistant was careful to avoid inflicting soft tissue injury due to traction, which may damage the mandibular facial nerve. After mandibular angle reduction was completed, the bony segment was gently removed (Figure 6). The divided masseter muscle was sutured together, and the skin flap was opened for drainage and sutured after 1 or 2 days. A gentle compression dressing was applied for one day. 

### 2.2. Statistical Analysis

We divided the patients into the intraoral and retroauricular incision groups. Postoperative results were assessed using preoperative and postoperative photographs. Furthermore, real operation time, global aesthetic improvement scale (GAIS) scores (Table 1) at 3 months postoperatively, and visual analog scale (VAS) scores on postoperative day 5 were compared. Statistical analyses were performed using SPSS ver. 12.0 for Windows (SPSS Corp., Armonk, NY, USA). Odds ratios were calculated using univariate and multivariate analyses. Statistical significance was set at *p* < 0.05.

## 3. Results

For each incision method, we managed 21 cases of mandibular angle reduction between April 2019 and October 2021. The average operation time for the retroauricular incision group (85 min) was relatively longer than that of the intraoral incision group (74 min); however, the difference was not statistically significant (*p* = 0.053). Surgery was successful in all the cases. An appropriate mandibular contour was achieved postoperatively (Figure 7 and Figure 8). All patients were satisfied with the surgical results regardless of the surgical method that they received. No significant difference was found in patient satisfaction according to the GAIS scores between the two groups (1.8 vs. 2.0, *p* = 0.379) (Table 2). There was no damage to the facial nerves, such as the mandibular nerve or parotid duct, during the operation. Massive bleeding occurred in one case due to blood vessel damage in the mouth incision; however, the bleeding was stopped by compression. Other minor complications were not significantly different between the two groups. Some patients experienced short-term complications, such as wound disruption of the incision line. However, these complications improved within 21 days after surgery. Follow-up periods ranged from 10 months to 2 years (with an average of 14 months). There were no serious long-term complications, such as facial palsy (Table 3). Compared to the intraoral incision, the pain felt by patients who underwent retroauricular incision was approximately 1.25 points lower on VAS on postoperative day 5 (*p* = 0.001) (Figure 9). In most patients, the condition of the scar was good without any special scar treatment, and there were no serious scar side effects, such as keloids (Figure 10). However, among the patients from the retroauricular approach group, two showed some hypertrophic scarring. These patients were treated with a single steroid injection and showed improvement (Figure 11).

## 4. Discussion

Several approaches have been introduced for cosmetic square-jaw resections. However, the most commonly used is the intraoral incision method. When the oral mucosa is incised, it is relatively safe in terms of avoiding damage to the facial nerve because dissection occurred through the deepest layer and scars are not visible. Additionally, since the oral mucosa can be submitted to a wide range of dissections, a wide field of view can be achieved. However, despite these advantages, some complications can occur during mandibular angle ostectomy using the intraoral approach [10,11]. Among these, massive bleeding due to blood vessel damage is the most serious and life-threatening complication.

While uncommon, the aforementioned side effect requires special attention because hemostasis is not easy when damage to large blood vessels, such as the facial artery or retromandibular vein, occurs. This complication may result from the fact that the mandible angle and posterior border of the ramus are not visible in the surgical field during intraoral access. Therefore, blind dissection is performed to detach the medial pterygoid muscle, and major blood vessels are torn when dissection is performed incorrectly. When such damage occurs, direct hemostasis is difficult because these blood vessels are not visible in the surgical field, resulting in massive bleeding and airway obstruction. Regarding minor complications and the postoperative recovery period of the patient, the intraoral approach often causes discomfort and long-term pain in the temporomandibular joint and lips due to excessive mouth opening during surgery. Additionally, since the mucous membrane is in a sutured state after incision and the drain is usually placed in the mouth, it is difficult to eat for 2–3 days after the removal of the drain. Even if the drain is quickly removed and soft food is eaten, it is very difficult to manage oral hygiene due to the sutures. To overcome these disadvantages of the intraoral approach, other approaches have been devised for mandibular angle resection. Lee et al. have successfully corrected a square jaw using a small incision in the submental area and a specially designed instrument, namely Lee’s rasp and saw [8]. However, this method has not been widely used because it requires a special instrument and much experience of the operation itself, owing to the nature of the blind operation. The transcutaneous approach associated with fracture reduction surgery for mandible angle fracture has been widely used for a long time [12]. Moreover, oral incision and skin incision are performed at the same time [13]. However, this incision method, also called the Risdon approach, is difficult to apply practically in cosmetic surgery, such as square jaw resection, due to visible scarring. 

To overcome the aforementioned disadvantages of the existing intraoral and transcutaneous incisions, we performed mandibular angle resection surgery using an incision behind the ear. We previously reported a square mandibular resection using an existing facelift incision in a patient group for whom facelift and mandibular angle reduction were performed together [9]. Mendelson et al. have published the anatomy of the pre-masseter space [14]. In the sub-SMAS plane, the premasseter space is located in the lateral lower third of the face between the platysma and masseteric fascia. Because the lower portion of the premasseter space is a safe plane, the risks of bleeding and facial nerve trauma can be reduced [9]. The marginal mandibular nerve exits the caudal border of the parotid gland, and it remains deep in the masseteric fascia. The point at which the facial artery crosses the lower border of the mandible is a reliable landmark for locating the marginal mandibular nerve [15]. Subplatysmal dissections closer to the mandibular angle and body should be performed carefully under direct vision. It maintains a more superficial dissection plane than the sub-SMAS fat, which can include nerve branches. For the mandibular nerve branch, blunt dissection is useful when dissecting beyond the parotid gland to avoid injury to the cervical branch [16]. According to the anatomic characteristics of the facial nerve, the authors performed a sub-SMAS dissection downward from the retroauricular incision to the parotid gland tail. In the medial direction, the surgical field was secured using the concept of the premasseter space, which is a safe dissection plane. The mandible angle was exposed by subperiosteal dissection, followed by the blunt dissection of the masseter fascia and muscle while examining the nerve distribution through direct vision. The posterior border of the mandibular angle is not visible during the intraoral approach but could be seen in the retroauricular approach as it comes into a relatively direct field of view, making it easier to determine the extent of bony resection. Additionally, it is possible to detach the medial pterygoid muscle attached to the remaining bone segment after resection, thereby reducing the side effects of vessel injury or bone segment entrapping. In this study, compared to the intraoral incision group, the retroauricular incision group did not have these side effects. Furthermore, because there was no need for the excessive opening of the mouth, mouth suturing, or drainage during surgery, patients were able to start eating relatively comfortably on day 1 after surgery, and the VAS score was significantly low (Figure 9). Typically, a conventional drain can cause additional scarring due to the need for puncture, and patients may find it difficult to manage drain tubes at home after discharge. There are also cases where drains accidentally detach during sleep or daily activities. In addition, when using a thin Penrose drain without a drain bag, it can be difficult to achieve adequate discharge to prevent hematoma. Therefore, in this study, we applied a compressive dressing that allows for natural drainage by leaving the skin wound open for 1–2 days, and then closed the wound once surgical site bleeding had stabilized. Unfortunately, there was some wound disruption behind the ear; however, these wounds healed with a simple dressing within 3 weeks. Although it was not possible to obtain accurate quantification, this surgery may have other advantages, such as the absence of contamination in the surgical site due to saliva and common flora in the oral cavity, as the incision was made through the skin and not through the gingiva. Consequently, we assumed that there would be fewer surgical site infections.

However, the retroauricular method had limitations. The resection of the angle is possible, but corticectomy to reduce the mandible width is difficult to perform. Therefore, it is not suitable for patients who want to alter their square face in the anterior profile. If genioplasty has to be performed simultaneously to improve the overall lower face contour, an incision in the mouth is required [17]. Additionally, the possibility of facial nerve damage exists if there is no clinical experience and understanding of accurate facial nerve anatomy (as detailed in the sections above). These disadvantages can be overcome through the precise measurement and planning of personalized pre-surgery through navigation and virtual reality images, which have recently attracted research interests, securing stable vision even in small incision spaces, and the diversification of dissection through robotic surgery, which have also received recent attention [18,19].

## 5. Conclusions

Mandible angle resection surgery using a retroauricular incision is a good surgical option for square-faced patients and is helpful for the quick recovery of the postoperative quality of life of patients.

## Figures and Tables

**Figure 1 jcm-12-02641-f001:**
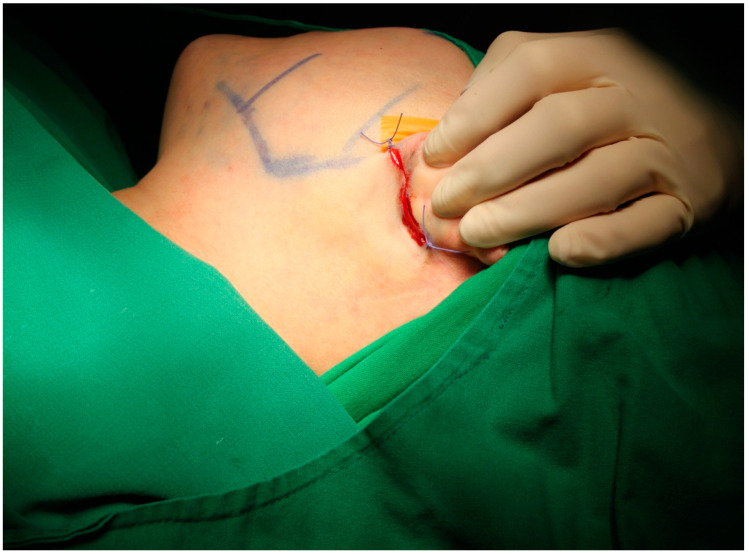
Preoperative bony landmark marking and retroauricular incision. By placing sutures at both ends of the incision, an unintended extension of the incision and skin damage due to traction could be prevented.

**Figure 2 jcm-12-02641-f002:**
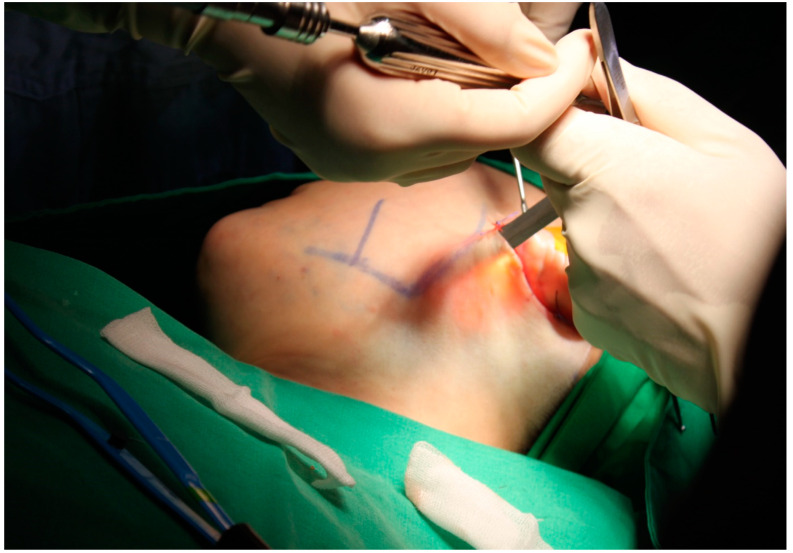
The superficial musculoaponeurotic system flap was lifted above the mandibular angular borderline and 2–3 cm below the earlobe. The contour of the lower parotid gland was examined to prevent iatrogenic damage.

**Figure 3 jcm-12-02641-f003:**
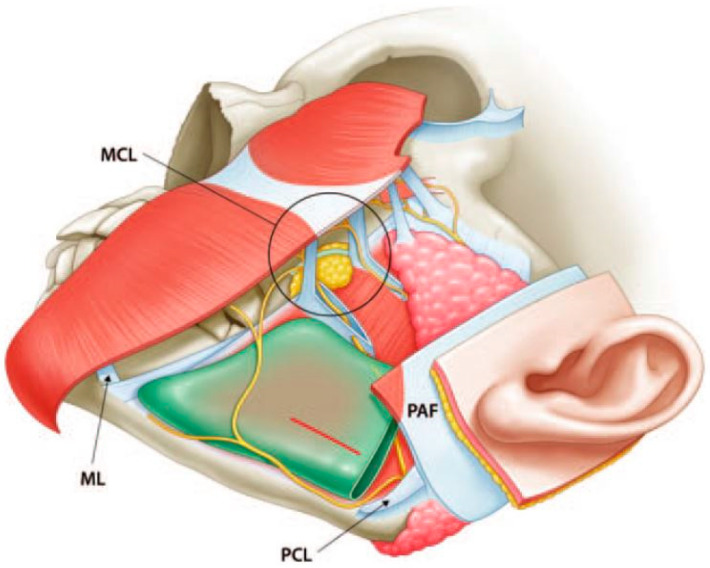
Anatomy of the premasseter space and facial nerve. The premasseter space is located at the lateral lower third of the face between the platysma and masseter fascia. The red horizontal line is the incision on the masseter fascia for subperiosteal dissection. The mandible angle can be approached by dissection on the masseter fascia and muscle between the buccal branch and the mandibular branch of the facial nerve. MCL, masseteric cutaneous ligament; ML, mandibular ligament; PAF, platysma auricular fascia; PCL, parotid cutaneous ligament.

**Figure 4 jcm-12-02641-f004:**
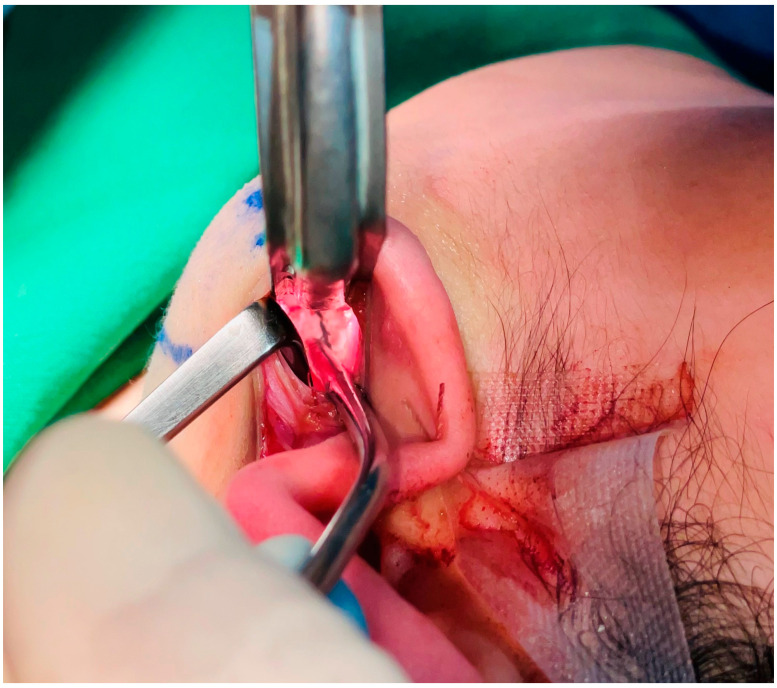
Exposure of the mandible angle secured after subperiosteal dissection. At this time, attention should be paid to nerve damage caused by excessive traction.

**Figure 5 jcm-12-02641-f005:**
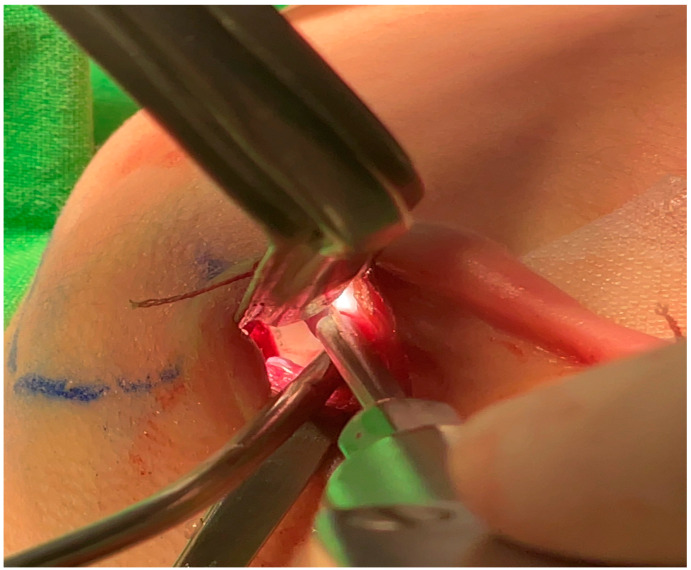
Sufficiently secured space within the ear incision and properly inserted saw.

**Figure 6 jcm-12-02641-f006:**
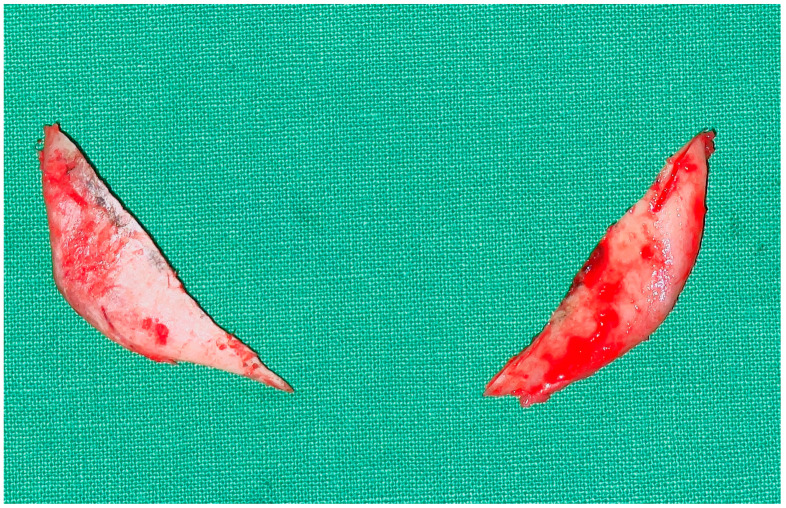
Bilateral bone fragment extruded after sufficient mandible angle ostectomy.

**Figure 7 jcm-12-02641-f007:**
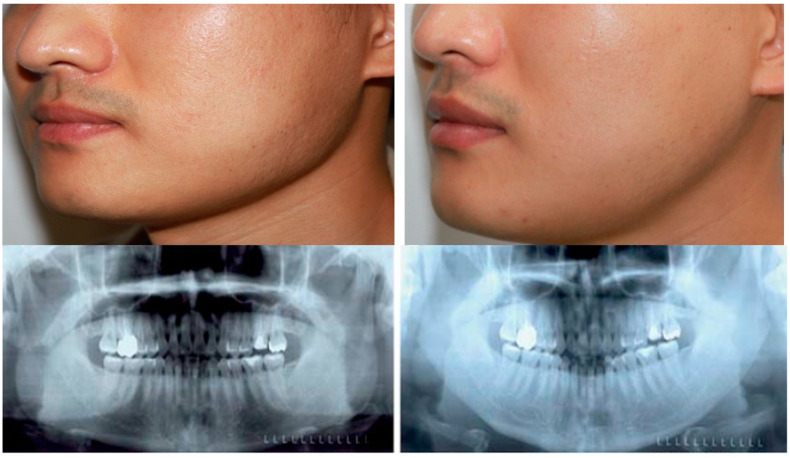
A 34-year-old woman underwent mandibular angle reduction via the retroauricular approach. (**Left**) Preoperative photograph and X-ray. (**Right**) Postoperative photograph and X-ray at 3 months after surgery, in which mandibular contour had significantly improved. Change from a rectangular face to an oval face could be observed in the postoperative photograph.

**Figure 8 jcm-12-02641-f008:**
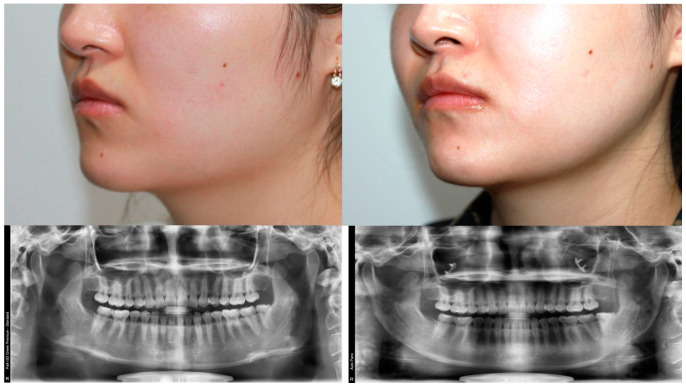
A 25-year-old woman with a prominent mandible angle underwent mandible angle reduction via the retroauricular approach. (**Left**) Preoperative photograph and X-ray. (**Right**) Postoperative photograph and X-ray at 3 months after surgery, in which the lower third facial contour was markedly slimmer.

**Figure 9 jcm-12-02641-f009:**
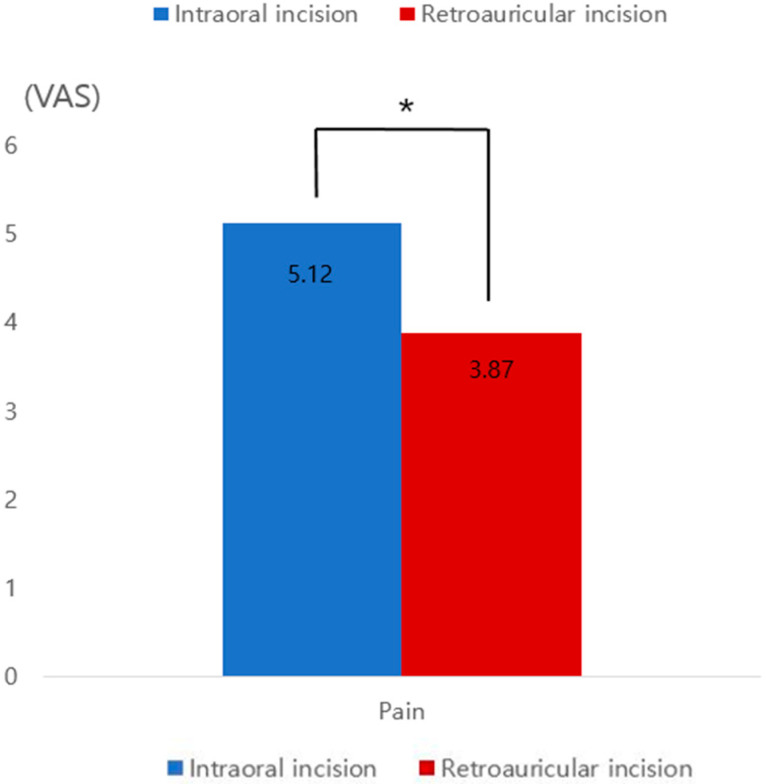
Visual analog scale (VAS) on postoperative day 5. Compared to the intraoral incision, the pain felt by the patient receiving the retroauricular incision was approximately 1.25 points lower. * A significant difference between the two groups (*p* < 0.01).

**Figure 10 jcm-12-02641-f010:**
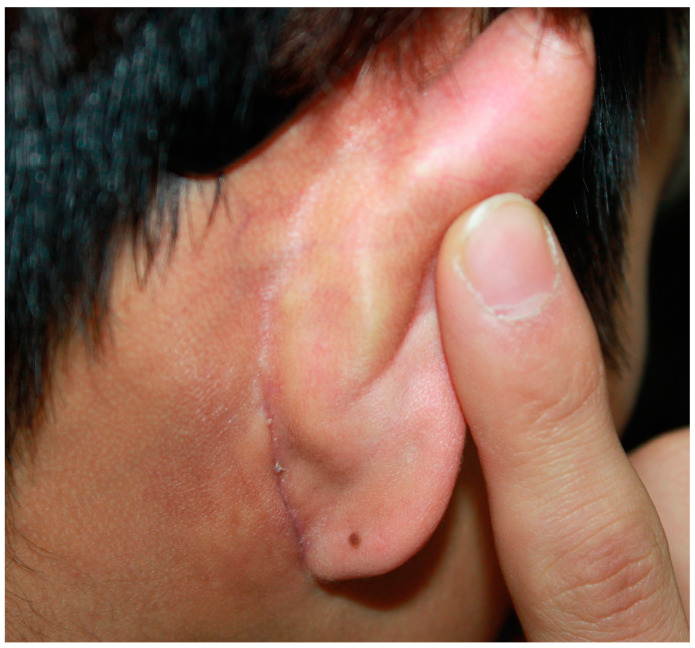
Postoperative photograph at 3 months after surgery of a 22-year-old man who underwent mandibular angle reduction via the retroauricular approach. The scar is nearly invisible.

**Figure 11 jcm-12-02641-f011:**
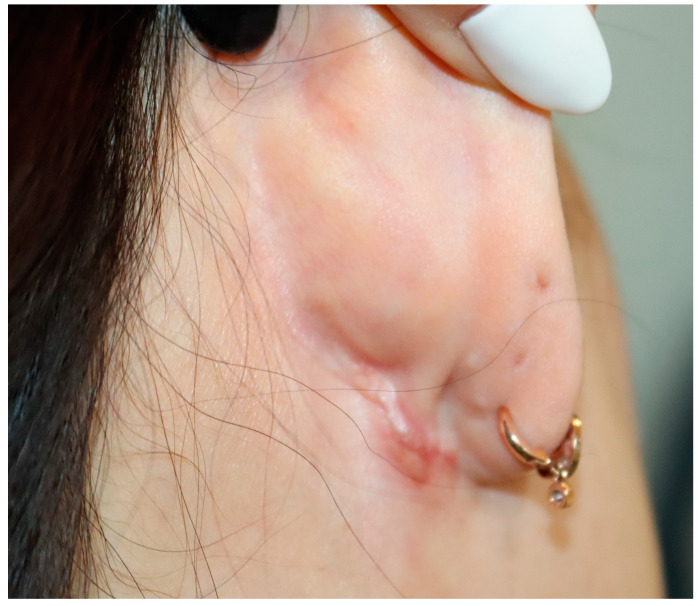
Postoperative photograph at 3 months after surgery of a 45-year-old woman who underwent mandibular angle reduction with the retroauricular approach. A hypertrophic scar occurred at the retroauricular incision line. As a treatment, a single steroid injection was administered and the scar improved.

**Table 1 jcm-12-02641-t001:** Global aesthetic improvement scale scores.

Degree	Description
1 Exceptional improvement	Excellent corrective result
2 Very improved patient	Marked improvement of the appearance, but not completely
3 Improved patient	Improvement of the appearance, better compared with the initial condition, but a touch-up is advised
4 Unaltered patient	The appearance substantially remains the same compared with the original condition
5 Worsened patient	The appearance has worsened compared with the original condition

**Table 2 jcm-12-02641-t002:** Comparison of the global aesthetic improvement scale scores (GAIS) between the intraoral incision and retroauricular incision.

Degree	A Group (Intraoral Incision) Number of Patients (%)	B Group (Retroauricular Incision) Number of Patients (%)
1 Exceptional improvement	7 (33.3)	5 (23.8)
2 Very improved patient	11 (52.4)	11 (52.4)
3 Improved patient	3 (14.2)	5 (23.8)
4 Unaltered patient	0	0
5 Worsened patient	0	0

**Table 3 jcm-12-02641-t003:** Complications of intraoral incision and retroauricular incision.

Complications	A Group (Intraoral Incision) Number of Patients (%)	B Group (Retroauricular Incision) Number of Patients (%)
Major complication		
Massive bleeding	1 (4.8)	0
Subcondyle fracture	0	0
Permanent facial palsy	0	0
Minor complication		
Numbness	2 (9.5)	3 (14.3)
Hematoma	3 (14.3)	2 (9.5)
Infection	2 (9.5)	1 (4.8)
Wound disruption	3 (14.3)	4 (19.0)

## Data Availability

The data presented in this study are available on request from the corresponding author. The data are not publicly available due to patients’ privacy.

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
