# Peer review of "Mandible Angle Resection with the Retroauricular Approach"

_jcm, 2023, doi:10.3390/jcm12072641_

Round 1

Reviewer 1 Report

Interesting contrasting view of mandible angle resection through a transcutaneous retroarticular approach. Few comments:

1) Line 216: The marginal mandibular nerve exits the caudal border of the parotid gland. And: remove period (one sentence)

2) Please show pictures of best and worst scars of this approach so readers can have an idea of such important outcomes associated with this technique.  With Asian skin, did authors notice any hypertrophic scars, keloids, or hyperpigmentation, and how to manage scar in this patient population. 

3) Was this a prospective or retrospective study in design? please clarify in methods. 

4) Authors should comment on any facial nerve injury or parotid injury in their experience 

5) Do authors use drain in the retro auricular approach or only compression dressing?

Author Response

Q1) Line 216: The marginal mandibular nerve exits the caudal border of the parotid gland. And: remove period (one sentence)

Answer 1) Thank you for your suggestion. We have made the necessary correction as you pointed out.

Q2) Please show pictures of best and worst scars of this approach so readers can have an idea of such important outcomes associated with this technique. With Asian skin, did authors notice any hypertrophic scars, keloids, or hyperpigmentation, and how to manage scar in this patient population.

Answer 2) We appreciate your feedback. We have added the information of scars to the results section on page 7, line 157-161, as well as representative figures (Figure 10, 11). The majority of patients who underwent surgery did not require special scar treatment due to the good condition of their scars, and that there were no serious scar side effects such as keloids. Additionally, there were no particular complaints about the location of the scars, as they were not visible on the front view. However, two patients showed some hypertrophic scar characteristics 2-3 months after surgery, and single steroid injection was administered, which resulted in improvement.

Q3) Was this a prospective or retrospective study in design? please clarify in methods.

Answer 3) Thank you for your comment. We have made the necessary revision to the material and methods section on page 2, line 68, to indicate that it was a prospective study.

Q4) Authors should comment on any facial nerve injury or parotid injury in their experience.

Answer 4) We have added a statement to the results section on page 7, line 148, indicating that there was no nerve or parotid duct damage.

Q5) Do authors use drain in the retro auricular approach or only compression dressing?

Answer 5) We would like to thank you for your insightful comments. Our research team believes that to use a drain, it requires additional puncture, which can cause scarring. Furthermore, managing the drain tube in clinical settings can be challenging, and there is a risk of unconscious removal during sleep. In addition, in the case of a thin Penrose drain without a drain bag, it may be difficult to achieve adequate discharge to prevent hematoma, so we adopted a compressive dressing to keep the skin wound open for 1-2 days to allow for natural drainage and then sutured the wound once it stabilized. We have added this information to the discussion section.

Reviewer 2 Report

This is a useful paper which just requires minor amendments.

The Abstract refers to 40 patients but the text refers to 42 ( 21 in each group ). Please correct one of these statements.

Introduction , P2, line 53 : Visual field restrictions from an intraoral approach - is this for the operator or the patient after surgery ? Yes I know what the authors intend to say, but it is not clear to read.

Line 96 - masseter , not master.

The retroauricular incision was left open for 1-3 days for drainage : why not place a small drain and close the skin ? Please explain.

Reference 1, from 1977, is concerned with bone surgery for masseteric hypertrophy - yet today we use botulinum toxin injections to shrink the masseter muscles, and often the bone will shrink as a result over time. Please update this part of the text and provide any more up to date references.

I thank the authors for an interesting study.

Author Response

Q1) The Abstract refers to 40 patients but the text refers to 42 ( 21 in each group ). Please correct one of these statements.

Answer 1) We would like to thank you for your insightful comments. As suggested, we have updated the number of participants mentioned in the abstract from 40 to 42, as the latter is the correct data based on our study.

Q2) Introduction , P2, line 53 : Visual field restrictions from an intraoral approach - is this for the operator or the patient after surgery ? Yes I know what the authors intend to say, but it is not clear to read.

Answer 2) We appreciate your feedback. We have revised the sentence to make it clearer that it meant the patient experiences significant pain and discomfort when the surgeon needs to examine the surgical site for dressing and follow-up observation after the surgery.

Q3) Line 96 - masseter , not master.

Answer 3) Thank you for your comment. We have made the necessary correction by changing "master" to the correct spelling of "masseter".

Q4) The retroauricular incision was left open for 1-3 days for drainage : why not place a small drain and close the skin ? Please explain.

Answer 4) Thank you for the considerate comment. Our research team believes that to use a drain, it requires additional puncture, which can cause scarring. Furthermore, managing the drain tube in clinical settings can be challenging, and there is a risk of unconscious removal during sleep. In addition, in the case of a thin Penrose drain without a drain bag, it may be difficult to achieve adequate discharge to prevent hematoma, so we adopted a compressive dressing to keep the skin wound open for 1-2 days to allow for natural drainage and then sutured the wound once it stabilized. We have added this information to the discussion section on page 12, line 253-260.

Q5) Reference 1, from 1977, is concerned with bone surgery for masseteric hypertrophy - yet today we use botulinum toxin injections to shrink the masseter muscles, and often the bone will shrink as a result over time. Please update this part of the text and provide any more up to date references.

Answer 5) Thank you for your valuable input. We have included additional information in the introduction section stating that there has been active use of botulinum toxin injections to correct hypertrophy of affected muscles in subsequent treatments, along with the relevant references.